# Oncolytic Adenovirus CD55-Smad4 Suppresses Cell Proliferation, Metastasis, and Tumor Stemness in Colorectal Cancer by Regulating Wnt/β-Catenin Signaling Pathway

**DOI:** 10.3390/biomedicines8120593

**Published:** 2020-12-11

**Authors:** Boduan Xiao, Leilei Zhang, Huihui Liu, Huiling Fang, Chunming Wang, Biao Huang, Xinyuan Liu, Xiumei Zhou, Yigang Wang

**Affiliations:** 1Xinyuan Institute of Medicine and Biotechnology, School of Life Sciences and Medicine, Zhejiang Sci-Tech University, Hangzhou 310018, China; 15067150683@163.com (B.X.); 201920201045@mails.zstu.edu.cn (L.Z.); 201920201010@mails.zstu.edu.cn (H.L.); fanghuiling2020@sibcb.ac.cn (H.F.); chunmingwang02@163.com (C.W.); jswxhb@163.com (B.H.); xyliu@sibcb.ac.cn (X.L.); zhouxiumei824@zstu.edu.cn (X.Z.); 2State Key Laboratory of Cell Biology, Institute of Biochemistry and Cell Biology, Shanghai Institutes for Biological Sciences, Chinese Academy of Sciences, Shanghai 200031, China

**Keywords:** oncolytic adenovirus, CD55-Smad4, cell proliferation, cell metastasis, tumor stemness, colorectal cancer, Wnt/β-catenin signaling pathway

## Abstract

During the past few decades, colorectal cancer (CRC) incidence and mortality have significantly increased, and CRC has become the leading cause of cancer-related death worldwide. Thus, exploring novel effective therapies for CRC is imperative. In this study, we investigated the effect of oncolytic adenovirus CD55-Smad4 on CRC cell growth. Cell viability assay, animal experiments, flow cytometric analysis, cell migration, and invasion assays, and Western blotting were used to detect the proliferation, apoptosis, migration, and invasion of CRC cells. The oncolytic adenovirus CD55-Smad4 was successfully constructed and effectively suppressed CRC cell proliferation in vivo and in vitro. Notably, CD55-Smad4 activated the caspase signaling pathway, inducing the apoptosis of CRC cells. Additionally, the generated oncolytic adenovirus significantly suppressed migration and invasion of CRC cells by overexpressing Smad4 and inhibiting Wnt/β-catenin/epithelial-mesenchymal transition (EMT) signaling pathway. Moreover, CRC cells treated with CD55-Smad4 formed less and smaller spheroid colonies in serum-free culture than cells in control groups, suggesting that CD55-Smad4 suppressed the stemness of CRC cells by inhibiting the Wnt/β-catenin pathway. Together, the results of this study provide valuable information for the development of a novel strategy for cancer-targeting gene-virotherapy and provide a deeper understanding of the critical significance of Smad4 in gene therapy of CRC.

## 1. Introduction

During the past few decades, the incidence and mortality of colorectal cancer (CRC) have significantly increased [1,2]. At present, CRC is the leading cause of cancer-related death worldwide. Thus, the study of novel and effective therapy methods for CRC is imperative.

Recently, oncolytic virotherapy became one of the most promising treatment strategies for solid malignancies [3,4,5,6]. Oncolytic viruses, such as oncolytic adenovirus (OA) [7,8], oncolytic vaccinia virus [9,10], and oncolytic herpes simplex virus [11,12] can specifically proliferate in tumor cells, resulting in their lysis. Cancer-targeting gene-virotherapy (CTGVT) proposed by our group a novel strategy to improve the anti-tumor effects of oncolytic viruses. CTGVT involves inserting an antitumor gene into an oncolytic viral vector (OA), such as oncolytic adenovirus. For example, Carcinoembryonic antigen (CEA)-controlled oncolytic adenovirus delivering TRAIL and MnSOD genes exert a more potent anti-tumor effect than the oncolytic virus alone [13]. Overexpression of tumor suppressor TSLC1 by a survivin-regulated OA significantly inhibits the growth of hepatocellular carcinoma [14]. A novel triple-regulated OA carrying the p53 gene exhibits potent anti-tumor efficacy in common human solid cancers [15]. Extensive research revealed that tumor-specific promoters could be used instead of the E1A promoter of OA to improve targeting. For example, a novel Golgi protein (GOLPH2)-regulated OA exhibits significant anti-tumor efficacy in hepatocellular carcinoma, liver cancer stem cells, and prostate cancer stem cells [7,8,16]. OA-mediated Hsp70 expression regulated by the CEA promoter has a strong anti-tumor efficacy in pancreatic cancer [17].

Mothers against decapentaplegic homolog 4 (Smad4) is a central critical signal transduction protein of the transforming growth factor-β (TGF-β) pathway and the bone morphogenetic protein (BMP-2) pathway. In many cancers, mutation or low expression of Smad4 switches its function from tumor inhibition to tumor promotion. For instance, miRNA-301a-3p promotes pancreatic cancer progression by downregulating Smad4 [18]. Overexpression of Smad4 suppresses cell growth and metastasis in CRC [19,20,21]. Thus, it may be more effective to use oncolytic viruses to carry the Smad4 gene in tumor therapy.

In this study, we used the CEA promoter to regulate the expression of E1A in oncolytic virus ZD55 and constructed a cancer-targeting gene-virus CD55-Smad4. Further, we confirmed that CD55-Smad4 has a potent anti-tumor efficacy in CRC in vitro and a mouse xenograft model. Notably, CD55-Smad4 effectively suppresses cell metastasis and tumor cell stemness in CRC by regulating the Wnt/β-catenin signaling pathway. Thus, the novel OA construct CD55-Smad4 represents a promising candidate drug for CRC therapy.

## 2. Materials and Methods

### 2.1. Cell and Culture

Human CRC cell lines HCT116, HT-29, SW620, and SW480 were obtained from the Type Culture Collection of the Chinese Academy of Sciences (Shanghai, China). All cells were cultured in DMEM (Gibco; Thermo Fisher Scientific, Inc., Waltham, MA, USA) supplemented with 10% fetal bovine serum (FBS; Gibco; Thermo Fisher Scientific, Inc.) and maintained at 37 °C in an atmosphere containing 5% CO_2_.

### 2.2. Crystal Violet Assay

The cells were seeded in 24-well plates (5 × 10^4^ cells/well), treated with CD55-EGFP and CD55-Smad4 at indicated MOIs (1, 5, 10, 20, and 40) for 48 h, and stained with 0.5% crystal violet solution. Subsequently, the cells were washed with distilled water and dried at 37 °C and photographed. Uninfected cells served as a control.

### 2.3. Cell Viability Assay

The cells were inoculated into 96-well plates at 5 × 10^3^ cell/well and 12 h later treated with CD55-EGFP and CD55-Smad4. After 24 h, 48 h, and 72 h, cell viability was examined by methyl thiazolyl tetrazolium (MTT; Sigma-Aldrich, St. Louis, MO, USA) assay. Briefly, 20 μL MTT (5 mg/mL in PBS) was added into each well, and after 4 h of incubation at 37 °C, the mediums and MTT were removed, and 150 μL of Dimethyl Sulfoxide was added. Samples were mixed thoroughly, and cell viability was assessed by measuring the absorbance at 490 nm using a microplate reader. Each experiment was repeated three times.

### 2.4. Hoechst 33342

A total of 5 × 10^4^ cells were inoculated into each well of a 24-well plate, cultured for 12 h, and treated with CD55-EGFP and CD55-Smad4 for an additional 48 h. Subsequently, the cells were incubated in Hoechst 33342 (Beyotime, Shanghai, China) for 10 min and washed twice with PBS. The cells were then observed under the IX71-22FL/PH fluorescence microscope (Olympus Corporation, Japan) at 200× magnification.

### 2.5. Western Blot Analysis

Western blot analysis was performed as previously described [13]. Briefly, the cells were lysed, protein concentration was detected by Pierce BCA protein assay kit (Thermo Fisher Scientific), and proteins were subjected to SDS–polyacrylamide gel electrophoresis. Separated proteins were transferred onto polyvinylidene difluoride membranes and incubated with primary antibodies against procaspase-3, cleaved caspase-3, PARP, β-catenin, E-cadherin, N-cadherin, vimentin, Oct4, Nanog, GAPDH (all from Cell Signaling Technology, Danvers, MA, USA), Smad4, CD44, and Sox2 (all from Santa Cruz Biotechnology, Santa Cruz, CA, USA).

### 2.6. Flow Cytometric Analysis

Cell apoptosis was determined using Annexin V FITC/PI (BD Biosciences, San Jose, CA, USA) using our previously described protocol [22]. In short, 5 × 10^5^ cells were inoculated into each well of 6-well plates, cultured for 12 h, and then treated with CD55-EGFP and CD55-Smad4 for an additional 48 h. Subsequently, the cells were harvested by trypsinization, resuspended in 500 µL of binding buffer, and stained with FITC-labeled annexin V and propidium iodide. The staining was evaluated immediately using the FACSCalibur flow cytometer (BD Biosciences).

PI/RNase Staining Buffer (BD Biosciences, San Jose, CA, USA) was used to determine the cell cycle distribution according to the manufacturer’s instructions. The cell cycle distribution was determined using the FACSCalibur flow cytometer (BD Biosciences). The fraction of cells in G0/G1, S, and G2/M phases was calculated and compared.

### 2.7. Cell Migration and Invasion Assays

Cell migration and cell invasion assays were performed according to previously described methods [23]. In brief, 24 well plates containing transwell chambers with polycarbonate membrane inserts (pore size: 8-µm; BD Biosciences) were used. For cell migration assay, 2 × 10^5^ cells treated with CD55-EGFP or CD55-Smad4 were seeded in each well in serum-free DMEM in the absence of Matrigel. For cell invasion assays, the same protocol was used, but the transwell chambers contained Matrigel (BD Biosciences). After 24 h of culture, the cells on the upper chamber were removed with a cotton swab, and the cells at the bottom of the membrane were fixed and stained with 0.5% crystal violet solution. The migrating and invading cells were counted in each well.

### 2.8. Spheroid Colony Formation Assay

Spheroid colony formation assay was performed as previously described [7]. Briefly, 1 × 10^3^ cells were plated in each well of ultra-low-attachment 6-well plates (Corning Life Sciences, Corning, NY, USA) in serum-free DMEM/F12 medium (Gibco) supplemented with 20 ng/mL recombinant human basic fibroblast growth factor (bFGF), 20 ng/mL recombinant human epidermal growth factor (EGF), 1 × B27, 100 IU/mL penicillin, and 100 μg/mL streptomycin. After culturing for 12 h, the cells were treated with CD55-EGFP and CD55-Smad4. After 1 week, spheroid colonies were counted in each well.

### 2.9. Animal Experiments

Animal experiments complied with the regulations and standards set by the US Department of Agriculture and the National Institutes of Health and were performed according to previously described protocols [22]. The study was approved by the Laboratory Animal Welfare Ethics Committee of the Zhejiang Sci-Tech University (No. 1909-23) in September 2019. Briefly, 4 weeks old female BALB/c nude mice were obtained from the Shanghai Experimental Animal Center (Shanghai, China). A total of 8 × 10^6^ HCT116 or HCT116-Smad4^−/−^ cells were subcutaneously inoculated into the right flank of the mice. When the xenograft tumors reached 90–130 mm^3^, the mice were divided randomly into 3 groups (8 mice per group) and injected twice, 24 h apart, with PBS (vehicle), CD55-EGFP (1 × 10^9^ PFU/mouse), and CD55-Smad4 (1 × 10^9^ PFU/mouse). After injecting the virus, tumor size was measured with a vernier caliper every five days and for a total of 30 days. Then, tumor tissue was harvested, fixed in 5% paraformaldehyde, dehydrated in a gradient of increasing ethanol concentrations, and embedded in paraffin. Five μm sections were cut and stained with hematoxylin and eosin for histological analysis.

### 2.10. Statistical Analysis

All data are presented as the mean ± standard deviation (SD). The differences were assessed by Student’s t-test and the analysis of variance (ANOVA) and were considered statistically significant at *p* < 0.05.

## 3. Results

### 3.1. Construction of Oncolytic Adenovirus CD55-Smad4

We have previously successfully constructed a CEA-controlled oncolytic adenovirus (CD55) [13]. CEA is a tumor marker with significantly increased expression in colon cancer [24,25]. Based on this construct, we generated a novel oncolytic adenovirus, CD55-Smad4, in which CD55 harbors the Smad4 gene (Figure 1A). CD55-EGFP (CD55 encoding the EGFP gene) was used as a control OA. PCR assay (Figure 1B) demonstrated the successful insertion of the Smad4 gene into CD55. Moreover, Western blot analysis (Figure 1C) indicated that HCT116 cells infected with CD55-Smad4 at the 1 MOI produced a significant amount of the transgene Smad4 protein.

### 3.2. Cytotoxic Effect of CD55-Smad4 and Inhibition of CRC Growth

To evaluate the anti-tumor effect of CD55-Smad4 in vitro, four CRC cell lines HCT116, HT-29, SW620, and SW480, were infected with various concentrations of CD55-Smad4 for 48 h. The crystal violet assay revealed that CD55-Smad4 provoked a greater cytopathic effect on CRC cells than CD55-EGFP (Figure 2A). The MTT assay further indicated that CD55-Smad4 had a greater inhibitory effect than CD55-EGFP (Figure 2B). Because the loss of the Smad4 gene could promote the initiation, development, migration, and invasion of CRC [26,27,28], and due to the high expression of Smad4 gene in HCT116 cells (Figure 2C) and the strongest cytotoxic effect of CD55-Smad4 on HCT116 cells (Figure 2B), for further experiments, the Smad4 gene was knocked down in HCT116 cells (HCT116-Smad4^−/−^) (Figure 2D). The MTT assay revealed that cell viability of HCT116 cells (Figure 2E) and HCT116-Smad4^−/−^ cells (Figure 2F) infected with CD55-Smad4 was significantly decreased in a time-dependent manner compared to cells infected with CD55-EGFP. Moreover, animal experiments showed that the growth of tumors in xenografted mice was more effectively inhibited in the groups treated with CD55-Smad4 than in groups treated with CD55-EGFP or PBS (Figure 2G,H). Hematoxylin and eosin (HE) staining indicated that CD55-Smad4 induced more severe cytopathic effects on tumor tissue than CD55-EGFP or PBS (Figure 2I,J). Thus, CD55-Smad4 has an enhanced ability to suppress cell proliferation in vivo and in vitro.

### 3.3. CD55-Smad4 Induced Cell Apoptosis and Its Mechanism in CRC Cells

To establish whether the inhibition of cell proliferation by CD55-Smad4 was due to the activation of apoptosis, this form of cell death was analyzed. The Hoechst assay showed that CD55-Smad4 triggered a higher level of nuclear fragmentation, the formation of apoptotic bodies, and chromatin condensation, inducing a higher magnitude of cell apoptosis than CD55-EGFP (Figure 3A,B). Additionally, flow cytometric analysis documented that the rate of cell apoptosis in the CD55-Smad4 group was markedly increased compared with the NC group or the CD55-EGFP group in HCT116 (Figure 3C) and HCT116-Smad4^−/−^ (Figure 3D) cells. Moreover, CD55-Smad4 induced the G2/M arrest in HCT116 (Figure 3E) or HCT116-Smad4^−/−^ (Figure 3F) cells more efficiently than NC or CD55-EGFP. To identify the mechanism of apoptosis activation, the caspase signaling pathway was analyzed by Western blotting in HCT116 or HCT116-Smad4^−/−^ cell. The results showed that CD55-Smad4 significantly activated caspase signaling and increased the cleavage of procaspase 3 and PARP, indicating that CD55-Smad4 can effectively induce cell apoptosis, enhancing the anti-tumor effect of CD55 (Figure 3G,H).

### 3.4. CD55-Smad4 Suppressed Metastasis and Cell Stemness in CRC by Regulating the Wnt/β-Catenin Signaling Pathway

Previous studies documented that the downregulation of the Smad4 gene promoted metastasis and cell stemness in tumors [26,29]. The cell migration (Figure 4A) and invasion (Figure 4B) assays performed in the present study revealed that CD55-Smad4 effectively suppresses CRC metastasis by overexpressing the Smad4 gene. Moreover, after growing under serum-free conditions for 1 week, HCT116 and HCT116-Smad4^−/−^ cells in the CD55-Smad4 group formed less and smaller spheroid colonies than those in the NC or CD55-EGFP group (Figure 4C). Mechanistically, the knockdown of Smad4 promoted the expression of β-catenin, while the treatment with CD55-Smad4 downregulated the expression of β-catenin, vimentin, and N-cadherin in HCT116 or HCT116-Smad4^−/−^ cells and upregulated expression of E-cadherin. These results suggest that CD55-Smad4 suppresses cell migration and invasion by inhibiting the Wnt/β-catenin signaling pathway involved in the epithelial-to-mesenchymal transition (EMT) (Figure 4D). Additionally, knockdown of Smad4 promoted the expression of stem cell markers CD44, Sox2, Oct4, and Nanog, while the treatment with CD55-Smad4 downregulated their expression in both HCT116 and HCT116-Smad4^−/−^ cells, implying that CD55-Smad4 suppresses CRC stemness by inhibiting the Wnt/β-catenin cascade (Figure 4E). Therefore, CD55-Smad4 suppressed metastasis and tumor cell stemness in CRC by regulating the Wnt/β-catenin signaling pathway.

## 4. Discussion

Oncolytic virus agents have become one of the most promising therapeutic strategies for solid malignancies [30,31,32]. Currently, more than 69 clinical trials for oncolytic virus alone or combination therapy in various solid tumors are registered in ClinicalTrials.gov. It is noteworthy that there are only two commercialized oncolytic virus agents: Oncorine (H101) approved by the Chinese SFDA in 2005 for nasopharyngeal carcinoma in combination with chemotherapy [33], and Talimogene laherparepvec (T-VEC) approved by the FDA in 2015 for the treatment of melanoma [34]. This progress highlights the hopes associated with the clinical value of oncolytic viruses, including modified adenovirus, vaccinia virus, herpes simplex virus, and reovirus. Oncolytic viruses can specifically proliferate in tumor cells and finally trigger their lysis without harming normal tissues [35,36,37]. In addition, cancer cells lysed by oncolytic viruses release tumor antigens that induce the immune response against the tumor. However, the tumor-killing efficacy of oncolytic viruses remains inadequate. Thus, the rational design and modification of oncolytic viruses are vital for their successful application.

Adenovirus is a versatile virus vector due to its handleability and the potential to become the oncolytic virus agent. The most common strategies for modifying the oncolytic adenovirus are substituting the endogenous E1A promoter with the tumor-specific promoter and insertion of an exogenous tumor-inhibitory gene [13]. For example, our previously constructed survivin-regulated oncolytic adenovirus carrying tumor suppressor TSLC1 significantly inhibits the growth of hepatocellular carcinoma [14]. Another novel Golgi protein (GOLPH2)-regulated oncolytic adenovirus GD55 exhibits potent anti-tumor efficacy in hepatocellular carcinoma and prostate cancer [7,8]. CEA is a good candidate for constructing oncolytic adenoviruses due to its high expression in colorectal cancer [38], pancreatic cancer [39] and. gastric cancer [40]. Thus, several recombinant oncolytic adenoviruses regulated by the CEA promoter and delivering tumor suppressor genes, such as ST13, TRAIL, and Mn-SOD, were constructed in our laboratory and exhibit potent activity against colorectal cancer, pancreatic ductal adenocarcinoma, and lung cancer [13,41]. These results suggest the clinical therapeutic value of CEA-controlled oncolytic adenoviruses.

Smad4 or DPC4 belongs to a family of signal transduction proteins, and inactivation and somatic mutations of Smad4 are often found in pancreatic [42], colorectal [43], and prostate cancers [44]. Thus, Smad4 acts as a tumor suppressor gene. In the current study, we successfully constructed a novel oncolytic adenovirus CD55-Smad4 containing the CEA promoter controlling the expression of E1A and delivering the Smad4 gene (Figure 1A). The CD55-Smad4 had a markedly higher cytopathic effect on CRC cells than CD55-EGFP and had a greater inhibitory effect on CRC cells than CD55-EGFP in vitro and in vivo (Figure 2). Additionally, CD55-Smad4 induced a higher level of cell apoptosis and G2/M arrest in CRC cells than CD55-EGFP (Figure 3A–F). The stimulation of apoptosis by CD55-Smad4 involved the activation of the caspase signaling pathway (Figure 3G,H). Thus, CD55-Smad4 is more effective in suppressing cell proliferation in vivo and in vitro, and more effectively activates the caspase signaling pathway to induce the cell apoptosis.

Smad4 is a central critical signal transduction protein of the TGF-β and BMP-2 pathway [45,46,47,48,49]. It was reported that the loss of Smad4 protein expression occurring in a subset of CRC patients is associated with poor prognosis [50]. Moreover, the restoration of Smad4 suppresses the Wnt/β-catenin signaling activity and ability to migrate in human colon carcinoma cells [51]. An earlier study pointed to the essential role of intracellular accumulation of β-catenin, the hallmark of Wnt signaling activation, in the origin of intestinal cancer stem cell and their invasive behavior [52]. The current work demonstrated that CD55-Smad4 could effectively suppress migration and invasion of CRC cells (Figure 4A,B). Moreover, knockdown of Smad4 promoted the expression of β-catenin, while the treatment with CD55-Smad4 downregulated the expression of β-catenin, vimentin, and N-cadherin in CRC cells, while the expression of E-cadherin was upregulated (Figure 4D). These findings indicate that CD55-Smad4 suppresses cell migration and invasion by inhibiting the Wnt/β-catenin/EMT signaling pathway. In addition, after growing under serum-free conditions for 1 week, CRC cells in the CD55-Smad4 group formed less and smaller spheroid colonies than in the NC group or CD55-EGFP groups (Figure 4C). Furthermore, knockdown of Smad4 promoted the expression of stem cell markers CD44, Sox2, Oct4, and Nanog in CRC cells, while the treatment with CD55-Smad4 downregulated their expressions (Figure 4E). This result implies that CD55-Smad4 suppresses the stemness of CRC cells by inhibiting the Wnt/β-catenin signaling pathway.

Although great progress in tumor treatment was achieved by using oncolytic adenoviruses [3,53,54], how to improve their therapeutic effects as a carrier of genes remains a clinically relevant question. Additionally, the heterogeneity of oncolytic adenoviruses may induce an inflammatory response, triggering an antiviral response of the immune system and reducing their bioavailability. Neutralizing antibody to the adenovirus present in the human organism significantly reduces the oncolytic adenovirus-mediated anti-tumor effect. Moreover, although the proliferation of the oncolytic adenovirus was inhibited in normal cells and tissues by deleting the adenoviral E1B 55-kDa gene, oncolytic viruses can continue to proliferate to some extent in certain normal cells and tissues [13]. This problem has to be considered when evaluating the safety of the clinical application of oncolytic viruses. Therefore, the application of oncolytic adenoviruses in cancer treatment requires additional in-depth studies, such as the possibility of structural modification of oncolytic adenoviruses [55,56] or their combination with nanomaterials [57,58]. In addition, the use of oncolytic adenoviruses together with chemotherapy provides more potent anti-tumor effects. For example, the combination of the oncolytic adenovirus ICOVIR-5 with chemotherapy produces an enhanced anti-glioma effect in vivo [59]. Oncolytic adenovirus and luteolin administered together exert synergistic anti-tumor effects against colorectal cancer cells and in a mouse model of CRC [22]. Oncolytic adenovirus carrying XAF1 and cisplatin synergistically suppress the growth of hepatocellular carcinoma [60]. In immunotherapy, the combination of immunogenic oncolytic adenovirus ONCOS-102 with anti-PD-1 pembrolizumab exhibits synergistic anti-tumor effect in the humanized A2058 melanoma huNOG mouse model [61].

Taken together, we successfully constructed a novel oncolytic adenovirus CD55-Smad4. The obtained results indicate that CD55-Smad4 suppresses cell proliferation, metastasis, and tumor stemness in CRC by regulating the Wnt/β-catenin signaling pathway (Figure 5). Therefore, this study provides valuable information for developing a novel strategy for cancer-targeting gene-virotherapy and advances our understanding of the critical role of Smad4 in gene therapy of CRC.

## Figures and Tables

**Figure 1 biomedicines-08-00593-f001:**
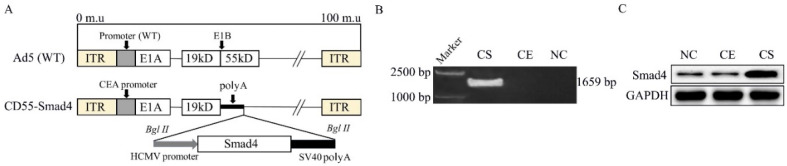
Construction of oncolytic adenovirus CD55-Smad4. (**A**) Schematic diagram of the structure of the recombinant oncolytic adenovirus structure. ITR, inverted terminal repeat. (**B**) PCR assay was used to detect the Smad4 gene. (**C**) Western blotting was performed to detect the expression of Smad4 protein. NC, negative control; CE, CD55-EGFP; CS, CD55-Smad4.

**Figure 2 biomedicines-08-00593-f002:**
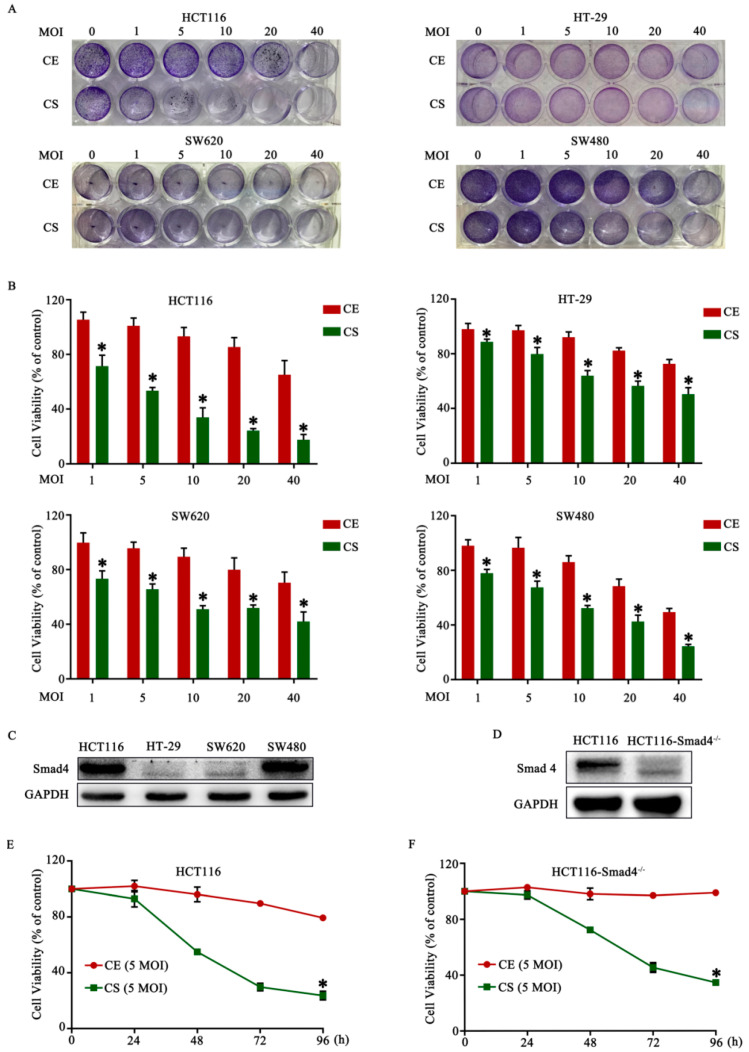
The cytotoxic effect of CD55-Smad4 and inhibition of colorectal cancer (CRC) growth. (**A**) Crystal violet assay was employed to determine the cytopathic effect of CD55-Smad4 in CRC cells. (**B**) MTT assay was used to detect the anti-tumor effect of CD55-Smad4 in CRC cells HCT116, HT-29, SW620, and SW480. (**C**) The expression of the Smad4 gene in CRC cells measured by Western blot assay. (**D**) The Smad4 gene was knocked down in HCT116 cells, and Western blotting was used to detect Smad4 expression in HCT116 and HCT116-Smad4^−/−^ cells. MTT assay was utilized to detect the anti-tumor effect of CD55-Smad4 in HCT116 (**E**) and HCT116-Smad4^−/−^ (**F**) cells. The volume of the tumor generated by HCT116 (**G**) and HCT116-Smad4^−/−^ (**H**) cells after the administration of PBS, CD55-EGFP, or CD55-Smad4. Hematoxylin and eosin staining of tumor formed by HCT116 (**I**) and HCT116-Smad4^−/−^ (**J**) cells. * *p* < 0.05. NC, Negative control; CE, CD55-EGFP; CS, CD55-Smad4; MOI, the multiplicity of infection.

**Figure 3 biomedicines-08-00593-f003:**
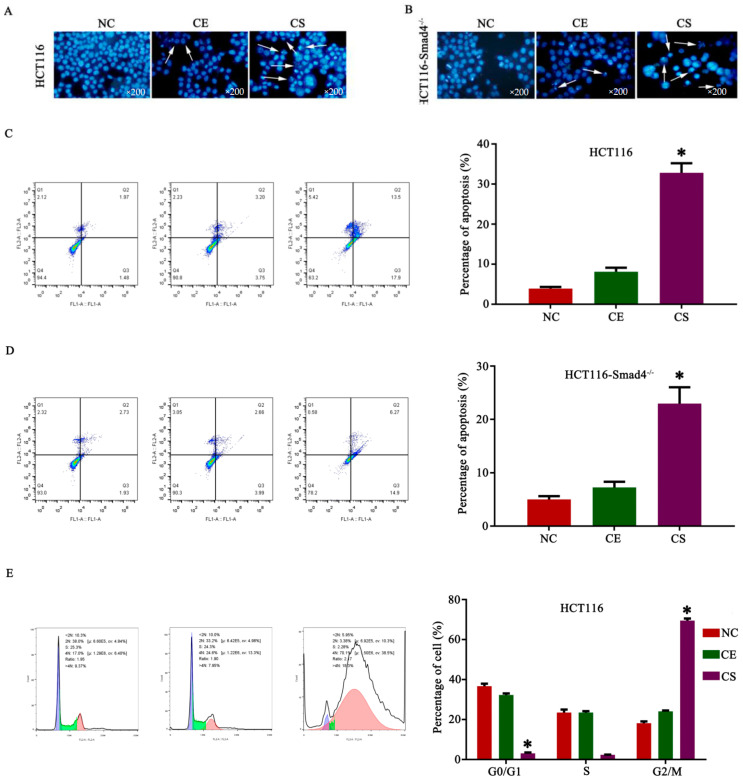
CD55-Smad4 induced cell apoptosis and its mechanism in CRC cells. A. The Hoechst assay in HCT116 (**A**) and HCT116-Smad4^−/−^ (**B**) cells after treatment with PBS, CD55-EGFP, or CD55-Smad4. Flow cytometric analysis of the percentage of apoptotic cells in HCT116 (**C**) and HCT116-Smad4^−/−^ (**D**) cells treated with PBS, CD55-EGFP, or CD55-Smad4. Flow cytometric analysis of the cell cycle in HCT116 (**E**) and HCT116-Smad4^−/−^ (**F**) cells after treatment with PBS, CD55-EGFP, or CD55-Smad4. Expression of apoptosis-associated proteins expression was detected by Western blotting in HCT116 (**G**) and HCT116-Smad4^−/−^ (**H**) cells. * *p* < 0.05. NC, Negative control; CE, CD55-EGFP; CS, CD55-Smad4.

**Figure 4 biomedicines-08-00593-f004:**
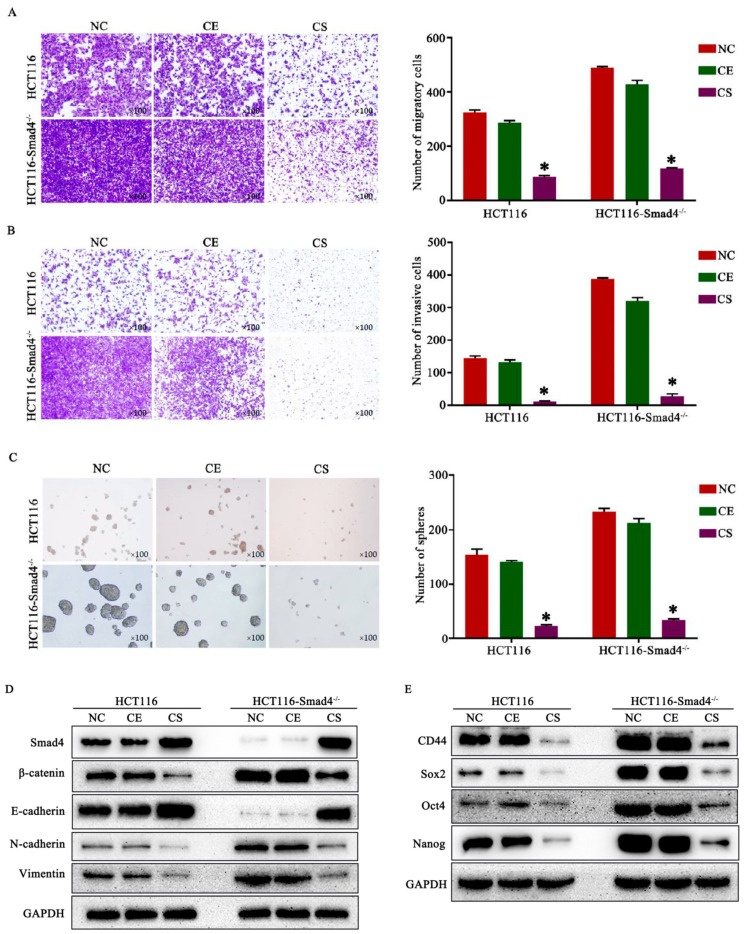
CD55-Smad4 suppressed metastasis and cell stemness in CRC by regulating the Wnt/β-catenin signaling pathway. Cell migration (**A**), invasion (**B**), and spheroid colonies (**C**) assay in HCT116 and HCT116-Smad4^−/−^ cell treated with PBS, CD55-EGFP, or CD55-Smad4. (**D**) The expression of proteins associated with the Wnt/β-catenin/EMT signaling pathway was evaluated by Western blotting. (**E**) Cancer stem cell markers were detected by Western blotting. * *p* < 0.05. NC, negative control; CE, CD55-EGFP; CS, CD55-Smad4.

**Figure 5 biomedicines-08-00593-f005:**
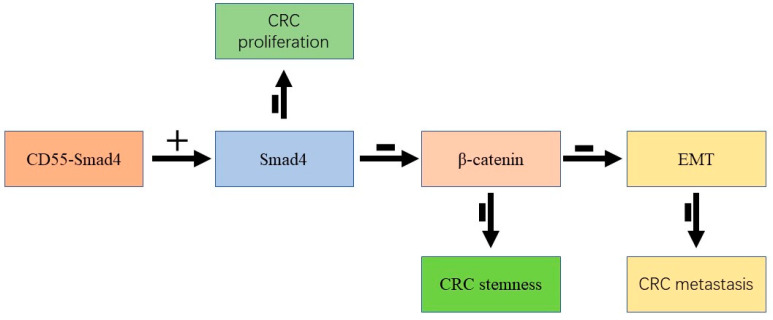
CD55-Smad4 suppressed cell proliferation, metastasis, and tumor stemness in CRC by regulating the Wnt/β-catenin signaling pathway. +: overexpression, -: inhibition.

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
