# Peer review of "Oncolytic Adenovirus CD55-Smad4 Suppresses Cell Proliferation, Metastasis, and Tumor Stemness in Colorectal Cancer by Regulating Wnt/β-Catenin Signaling Pathway"

_biomedicines, 2020, doi:10.3390/biomedicines8120593_

Round 1
Reviewer 1 Report
I have no suggestion to improve the quality of the revised version of the manuscript. The manuscript is suitable for publication.
Reviewer 2 Report
Minor comments:
1) Please correct as "spheroid colony formation assay"
2) Figure 2 legend: Please add A. and B. in parentheses, as in (C), (D), etc.
This manuscript is a resubmission of an earlier submission. The following is a list of the peer review reports and author responses from that submission.
Round 1
Reviewer 1 Report
In this manuscript Boduan Xiao, et al. described results of a study in the field of oncolytic viral therapy. For this study they constructed a new type of oncolytic adenovirus and investigated different effects of the construction using several in vitro and in vivo models of colorectal cancer. The authors revealed significant antitumor effects of the developed preparation. The study demonstrated that the mechanisms of anticancer effects of the created gene-viro construction was connected with induction of tumor cell apoptosis, inhibition of metastasis and tumor stemness. The obtained results suggest that described construction may be considered as a new effective preparation for anticancer virotherapy.
In general the study is well presented, proper controls are used and the conclusions are convincingly supported by experimental results, the data of considerable novelty and interest. I have some minor comments:
- There is no description of “The crystal violet assay” (line 77-78, Figure 2) in Materials and Methods.
- There is confusion in Figure 2 legend for G, H, I and J. Here also authors should decode the abbreviation MOI.
- As a part of discussion the authors should mention possible interaction of the developed viral construction with normal cells and tissues.
- Authors should correct misprints in the text (e.g. “ZD55” – line 57).
Author Response
Dear reviewer,
Firstly, we should express our sincere thanks to your seriously review for our manuscript. Thank you for your suggestions and comments. According to your suggestions, we made an earnest modification throughout the manuscript. Please check the revision of this manuscript.
Reviewer 1:
In this manuscript Boduan Xiao, et al. described results of a study in the field of oncolytic viral therapy. For this study they constructed a new type of oncolytic adenovirus and investigated different effects of the construction using several in vitro and in vivo models of colorectal cancer. The authors revealed significant antitumor effects of the developed preparation. The study demonstrated that the mechanisms of anticancer effects of the created gene-viro construction was connected with induction of tumor cell apoptosis, inhibition of metastasis and tumor stemness. The obtained results suggest that described construction may be considered as a new effective preparation for anticancer virotherapy.
In general the study is well presented, proper controls are used and the conclusions are convincingly supported by experimental results, the data of considerable novelty and interest. I have some minor comments:
Point 1: There is no description of “The crystal violet assay” (line 77-78, Figure 2) in Materials and Methods.
Response 1: Thanks for your kindly suggestion and comments. We have described “The crystal violet assay” in Materials and Methods. Please see line 68-72. (2.2. The crystal violet assay: The cells (50, 000) were seeded in 24-well plates for each well and treated with CD55-EGFP and CD55-Smad4 of indicated MOIs (1, 5, 10, 20 and 40) for 48h, stained with 0.5% crystal violet solution, then washed with distilled water and dried at 37℃,. and documented by photography. Uninfected cells served as control).
Point 2: There is confusion in Figure 2 legend for G, H, I and J. Here also authors should decode the abbreviation MOI.
Response 2: Thanks for your remind. We have corrected Figure 2 legend for G, H, I and J. We also decode the abbreviation MOI (multiplicity of infection).
Point 3: As a part of discussion the authors should mention possible interaction of the developed viral construction with normal cells and tissues.
Response 3: Thanks for your suggestion. According to your suggestion, we added possible interaction of the developed viral construction with normal cells and tissues in the discussion in Line 245-248 (Moreover, although the proliferation of oncolytic adenovirus was inhibited in normal cells and tissues by deleting adenovirus E1B 55-kDa gene, in fact, oncolytic viruses can still proliferate with a small amount in some normal cells and tissues, thus it is worth considering about safety issues).
Point 4: Authors should correct misprints in the text (e.g. “ZD55” – line 57).
Response 4: Thanks a lot. We correct misprints in the text in Line 58. In this manuscript, CD55 is constructed by using the CEA promoter instead of the E1A promoter in ZD55.
Reviewer 2:
Point 1: Several sentences need rephrasing or are completely wrong (e.g., "Thus, the novel and effective therapy methods for CRC is imperative"; "...we investigated the effect of oncolytic adenovirus CD55-Smad4 in suppresses CRC cell growth", "animal experiment was used", "...CD55-Smad4 can effectively suppressed cell metastasis", "Thus, we constructed CD55-Smad4 that were CD55 carrying Smad4 gene" and the list goes on).
Response 1: Thanks for your kind remind and comments. We have corrected the above language mistakes, and checked and modified the language through the manuscript. Please check the modified manuscript.
Point 2: Plenty of grammatical errors ("oncolytic virotherpay", etc). In Fig. 1A, please replace "ployA" with "poly A", and in Fig. 1B, "Maker" with "Marker". In Fig. 2, please replace "Samd4" with "Smad4".
Response 2: Thanks for your help to point out many grammatical errors. Other than the above errors you mentioned, we also checked and modified all other errors throughout the manuscript. Please check the modified manuscript. Thanks again for improving our manuscript.
Point 3: Why was the adenovirus termed "CD55-Smad4" and not ZD55-Smad4? Perhaps it would make more sense since the oncolytic adenoviral system with E1B 55kD gene deletion (ZD55) was used to construct it.
Response 3: Thanks for your suggestion. Oncolytic adenovirus ZD55-gene system was first designed and constructed in our group through E1B 55kD gene deletion (ZD55). We used ZD55-gene system to deliver a series of antitumor gene and obtained their recombinant oncolytic adenovirus. But in this manuscript, we used CD55 as the oncolytic adenovirus vector because CD55 is constructed by using the CEA promoter instead of the E1A promoter in ZD55, and other construct in both CD55 and ZD55 is same. The use of CEA promoter can improve targeting ability of oncolytic adenovirus in CEA positive CRC cells. Thus, we constructed and termed CD55-Smad4 through CD55 carrying Smad4 gene.
Point 4: Please define CEA (carcinoembryonic antigen) and all other acronyms once first appearing in the text.
Response 4: Thanks for your suggestions. According to your remind, we defined (carcinoembryonic antigen) and all other acronyms once first appearing in the manuscript. Please check the modified text.
Point 5: Figures 2 and 3 are small and of really poor quality. Please fix them.
Response 5: Thanks for your remind. Actually, the Figures in the manuscript are compressed, and the original Figures are clear and high quality. It need to upload after further procession for the manuscript.
Point 6: The number of references is very small, just 33, a good portion of which do not ever focus on colorectal cancer.
Response 6: Thanks for your suggestion. According to your suggestion, we have added some references about colon cancer. Please check the revised manuscript.
Reviewer 2 Report
1) Several sentences need rephrasing or are completely wrong (e.g., "Thus, the novel and effective therapy methods for CRC is imperative"; "...we investigated the effect of oncolytic adenovirus CD55-Smad4 in suppresses CRC cell growth", "animal experiment was used", "...CD55-Smad4 can effectively suppressed cell metastasis", "Thus, we constructed CD55-Smad4 that were CD55 carrying Smad4 gene" and the list goes on).
2) Plenty of grammatical errors ("oncolytic virotherpay", etc). In Fig. 1A, please replace "ployA" with "poly A", and in Fig. 1B, "Maker" with "Marker". In Fig. 2, please replace "Samd4" with "Smad4".
3) Why was the adenovirus termed "CD55-Smad4" and not ZD55-Smad4? Perhaps it would make more sense since the oncolytic adenoviral system with E1B 55kD gene deletion (ZD55) was used to construct it.
3) Please define CEA (carcinoembryonic antigen) and all other acronyms once first appearing in the text.
4) Figures 2 and 3 are small and of really poor quality. Please fix them.
5) The number of references is very small, just 33, a good portion of which do not ever focus on colorectal cancer.
Author Response
Dear reviewer,
Firstly, we should express our sincere thanks to your seriously review for our manuscript. Thank you for your suggestions and comments. According to your suggestions, we made an earnest modification throughout the manuscript. Please check the revision of this manuscript.
Reviewer 2:
Point 1: Several sentences need rephrasing or are completely wrong (e.g., "Thus, the novel and effective therapy methods for CRC is imperative"; "...we investigated the effect of oncolytic adenovirus CD55-Smad4 in suppresses CRC cell growth", "animal experiment was used", "...CD55-Smad4 can effectively suppressed cell metastasis", "Thus, we constructed CD55-Smad4 that were CD55 carrying Smad4 gene" and the list goes on).
Response 1: Thanks for your kind remind and comments. We have corrected the above language mistakes, and checked and modified the language through the manuscript. Please check the modified manuscript.
Point 2: Plenty of grammatical errors ("oncolytic virotherpay", etc). In Fig. 1A, please replace "ployA" with "poly A", and in Fig. 1B, "Maker" with "Marker". In Fig. 2, please replace "Samd4" with "Smad4".
Response 2: Thanks for your help to point out many grammatical errors. Other than the above errors you mentioned, we also checked and modified all other errors throughout the manuscript. Please check the modified manuscript. Thanks again for improving our manuscript.
Point 3: Why was the adenovirus termed "CD55-Smad4" and not ZD55-Smad4? Perhaps it would make more sense since the oncolytic adenoviral system with E1B 55kD gene deletion (ZD55) was used to construct it.
Response 3: Thanks for your suggestion. Oncolytic adenovirus ZD55-gene system was first designed and constructed in our group through E1B 55kD gene deletion (ZD55). We used ZD55-gene system to deliver a series of antitumor gene and obtained their recombinant oncolytic adenovirus. But in this manuscript, we used CD55 as the oncolytic adenovirus vector because CD55 is constructed by using the CEA promoter instead of the E1A promoter in ZD55, and other construct in both CD55 and ZD55 is same. The use of CEA promoter can improve targeting ability of oncolytic adenovirus in CEA positive CRC cells. Thus, we constructed and termed CD55-Smad4 through CD55 carrying Smad4 gene.
Point 4: Please define CEA (carcinoembryonic antigen) and all other acronyms once first appearing in the text.
Response 4: Thanks for your suggestions. According to your remind, we defined (carcinoembryonic antigen) and all other acronyms once first appearing in the manuscript. Please check the modified text.
Point 5: Figures 2 and 3 are small and of really poor quality. Please fix them.
Response 5: Thanks for your remind. Actually, the Figures in the manuscript are compressed, and the original Figures are clear and high quality. It need to upload after further procession for the manuscript.
Point 6: The number of references is very small, just 33, a good portion of which do not ever focus on colorectal cancer.
Response 6: Thanks for your suggestion. According to your suggestion, we have added some references about colon cancer. Please check the revised manuscript.
Round 2
Reviewer 2 Report
The authors have put an effort to ameliorate their manuscript based on my suggestions; however, I would still like to see a more thorough Discussion. Regarding the references, they have just inserted two new citations, but I think that the number could be increased more, along with the Discussion section.